# Applications of Artificial Intelligence for Metastatic Gastrointestinal Cancer: A Systematic Literature Review

**DOI:** 10.3390/cancers17030558

**Published:** 2025-02-06

**Authors:** Amin Naemi, Ashkan Tashk, Amir Sorayaie Azar, Tahereh Samimi, Ghanbar Tavassoli, Anita Bagherzadeh Mohasefi, Elaheh Nasiri Khanshan, Mehrdad Heshmat Najafabad, Vafa Tarighi, Uffe Kock Wiil, Jamshid Bagherzadeh Mohasefi, Habibollah Pirnejad, Zahra Niazkhani

**Affiliations:** 1Nordcee, Department of Biology, University of Southern Denmark, 5230 Odense, Denmark; 2Cognitive Systems, DTU Compute, The Technical University of Denmark (DTU), 2800 Copenhagen, Denmark; ashta@dtu.dk; 3SDU Health Informatics and Technology, The Maersk Mc-Kinney Moller Institute, University of Southern Denmark, 5230 Odense, Denmark; aaz@mmmi.sdu.dk (A.S.A.); ukwiil@mmmi.sdu.dk (U.K.W.); j.bagherzadeh@urmia.ac.ir (J.B.M.); 4Department of Computer Engineering, Urmia University, Urmia 165, Iran; anita.bagher@gmail.com (A.B.M.); elahe.nasiri044@gmail.com (E.N.K.); mehrddy@gmail.com (M.H.N.); vafatarighi1379@gmail.com (V.T.); 5Student Research Committee, Urmia University of Medical Sciences, Urmia 1138, Iran; tsamimi66@gmail.com; 6Department of Medical Informatics, Urmia University of Medical Sciences, Urmia 1138, Iran; 7Department of Computer Engineering, Urmia Branch, Islamic Azad University, Urmia 969, Iran; gh.tavassoli@gmail.com; 8Patient Safety Research Center, Clinical Research Institute, Urmia University of Medical Sciences, Urmia 1138, Iran; h.pirnejad@amsterdamumc.nl; 9Department of Family Medicine, Amsterdam University Medical Center, 7057 Amsterdam, The Netherlands; 10Nephrology and Kidney Transplant Research Center, Clinical Research Institute, Urmia University of Medical Sciences, Urmia 1138, Iran; niazkhani@eshpm.eur.nl; 11Erasmus School of Health Policy and Management (ESHPM), Erasmus University Rotterdam, 3000 Rotterdam, The Netherlands

**Keywords:** artificial intelligence, gastrointestinal cancer, metastasis, systematic review

## Abstract

This research investigates the use of Artificial Intelligence (AI) in improving the diagnosis, treatment, and follow-up of metastatic gastrointestinal cancers. By analyzing studies published between 2010 and 2022, the research evaluates the potential of AI models in enhancing diagnostic accuracy, predicting treatment outcomes, and identifying biomarkers. The findings highlight the promise of AI, particularly machine learning and deep learning, in advancing clinical practice. However, the study also identifies challenges, such as the reliance on retrospective data, inconsistencies in imaging protocols, small sample sizes, and issues related to data preprocessing and model interpretability. These challenges hinder the broad clinical implementation of AI models. This research aims to inform future studies and guide the integration of AI into clinical settings, with the goal of improving patient outcomes and streamlining treatment strategies for gastrointestinal cancers.

## 1. Introduction

Gastrointestinal cancers, particularly gastric and colorectal cancers, are among the most prevalent and deadly malignancies worldwide. These cancers significantly contribute to the global cancer burden, with colorectal cancer being the third most common cancer and the second leading cause of cancer-related mortality in industrialized and developing countries [1]. Gastric cancer, despite its declining incidence in some regions, remains a leading cause of cancer death globally [1,2]. The prognosis for advanced stages of these cancers remains poor, with metastasis being a critical factor contributing to the high mortality rates. Specifically, liver metastases are common in colorectal cancer, affecting approximately 50% of patients [3]. Similarly, gastric cancer frequently metastasizes to the liver and other distant organs [1]. Therefore, early detection, accurate diagnosis, and effective treatment of metastases are crucial to improve patient outcomes.

The management and study of gastrointestinal cancers rely heavily on comprehensive data collection. Data routinely collected from patients include demographic data (e.g., age and sex), tumor characteristics (e.g., size, location, stage, and histopathological findings), and treatment history (e.g., surgery, chemotherapy, and radiotherapy). For instance, studies often report the stage of cancers, which is critical for determining the extent of disease spread and for planning treatment [4,5,6]. Imaging data, particularly from Computed Tomography (CT) and Magnetic Resonance Imaging (MRI), play a pivotal role in diagnosing and monitoring these cancers. Radiomics features, which are quantitative features extracted from medical images, have gained prominence. These features include measures of texture, shape, and intensity that capture the heterogeneity of tumors and surrounding tissues [7,8,9]. Additionally, laboratory data such as Carcinoembryonic Antigen (CEA) levels, which are biomarkers for colorectal cancer, are frequently collected to monitor disease progression and response to treatment [10,11].

Artificial Intelligence (AI) has emerged as a transformative tool in the field of medical imaging and oncology, offering the potential to enhance the accuracy and efficiency of cancer diagnosis and treatment planning. AI algorithms, particularly machine learning (ML) and deep learning (DL) models, are adept at analyzing complex datasets and identifying patterns that may not be apparent to human observers [12]. In the context of gastrointestinal cancers, AI has been applied to various tasks, including detecting metastases, predicting treatment outcomes, and identifying prognostic biomarkers.

Several studies have demonstrated the effectiveness of AI in improving diagnostic performance [1,2,3,4,13]. For example, DL models, such as Convolutional Neural Networks (CNNs) [12], have been used to analyze CT images for the detection of extranodal extension in cervical lymph node metastases, achieving higher accuracy than traditional radiologist evaluations [2,4]. Similarly, ML-based radiomics models have been developed to predict local tumor progression and the occurrence of metachronous liver metastases in colorectal cancer patients [1,3]. These models combine radiomics features with clinical data to provide a comprehensive assessment of disease risk, facilitating personalized treatment planning.

While AI has shown promise results in various medical fields, its application in metastatic gastrointestinal cancers remains relatively unexplored and is evolving rapidly. Given the complexity and heterogeneity of these cancers, as well as the critical importance of timely and precise interventions, it is essential to assess the current state of AI-driven approaches in this area. This review seeks to address this gap by systematically evaluating existing research on AI in metastatic gastrointestinal cancers. By synthesizing the current evidence, the review will provide a comprehensive overview of AI’s effectiveness, identify key challenges, and highlight areas that require further exploration. In carrying this out, it will not only contribute to a deeper understanding of AI’s potential in this domain but also help guide future research and clinical practices. The findings of this review will offer valuable insights for clinicians and researchers, helping to optimize AI applications, inform new strategies, and ultimately improve patient outcomes in the management of metastatic gastrointestinal cancers.

## 2. Materials and Methods

This review covered the period from January 2010 to January 2022 and focused on the questions listed in Table 1.

### 2.1. Search Strategy and Study Selection

We searched four databases: PubMed, Scopus, Embase (Ovid), and Google Scholar. We used a variety of keywords and criteria grouped into five categories: ML keywords, medical keywords, document type, publication year, and language. Criteria within each group were linked with OR operators, and all groups were linked with AND operators. Table 2 displays the search keywords applied to titles, abstracts, and full texts across the three databases. The inclusion and exclusion criteria are provided in Table 3. The full keywords for different databases are presented as a Appendix A.

### 2.2. Data Extraction

Three researchers (AN, AT, and ASA) individually reviewed titles and abstracts, utilizing the Covidence tool. Following this step, ten researchers (AN, AT, ASA, ABM, ENK, MH, VT, TS, GT, and ZN) as AI and medical teams, respectively, performed full-text reviewing. AN and AT evaluated the extracted data for the AI team, and ZN and HP assessed the extracted data for the clinical team. Spreadsheets for data extraction were organized according to the Critical Appraisal and Data Extraction for Systematic Reviews (CHARMS) checklist for systematic reviews [13]. A spreadsheet of the checklist is provided as a Appendix A.

### 2.3. Risk of Bias Assessment

The risk of bias (ROB) for each study was evaluated using the prediction risk of bias assessment tool (PROBAST) checklist and reported based on a modified version [14]. Extracted articles were categorized as low, high, or unclear risk. A study was classified as having a high ROB if it exhibited high risk in at least one of four domains (participants, predictors, outcome, and analysis). Three researchers (AN, AT, and ASA) conducted the ROB assessment and it is provided as a Appendix A.

## 3. Results

The initial search yielded 5123 records, which were then subjected to duplicate removal, eligibility assessment, full-text evaluation, quality appraisal, and review of references. These processes led to the identification of 46 final original studies specifically focused on AI applications in metastatic gastrointestinal cancers. Besides these 46 studies, the original study list covered a wide range of other cancers, including breast cancer, melanoma and other skin cancers, lung cancer, genitourinary cancers (comprising prostate, bladder, testicular, and kidney cancers), sarcoma, female reproductive tract cancers (including cervical, ovarian, and endometrial cancers), hematological cancers (such as lymphoma and leukemia), head and neck cancers, brain cancers, and thyroid cancers (listed in descending order of frequency). Figure 1 illustrates a flowchart outlining the study selection process. 

### 3.1. Patient Types and Data Collection Period

Twenty-four studies investigated colorectal cancer patients. Nine studies included patients with gastric cancer, while seven studies focused on cancers in the oral cavity, two focused on the esophagus, two on the liver, one on the pancreas, and one on cancer of undocumented gastrointestinal origin.

The data collection period in the included studies ranged from a minimum of 8 months to a maximum of 180 months (mean = 45.81). This information was missing in six studies. The range of included patients varied from a minimum of 19 to a maximum of 116,878 individuals. As seen in Table 4, more than half of the studies used fewer than 200 patients for their model development.

### 3.2. Age and Gender Distribution

All extracted data from the final set of included studies are presented in Table 5 and Table 6. The included studies reported patient age primarily by mean values, with reported ages ranging from 52 to 74 years. Some studies used median values to represent age. The “Male” column reflects the percentage of male participants, which varies from approximately 49% to 80%, indicating a predominantly male sample in most studies. Several studies [15,16] reported male dominance with percentages over 70%. The variability in reporting and occasional gaps in demographic data, such as missing age or gender information, could affect the interpretation and generalizability of the study results across broader patient populations. The disparity in gender representation may reflect the gender-specific prevalence of certain types of gastrointestinal cancers or the selection criteria of the studies. This variability and occasional lack of comprehensive reporting for these two features can impact the interpretation and generalizability of the study results to broader patient populations.

### 3.3. Datasets and Features

In most studies, data were collected by the authors themselves from specific hospitals or cancer centers, while a few studies relied on publicly available datasets for model development. In total, 32 studies used data collected by the authors, and 13 studies used public data sources (with 2 of them using the Surveillance, Epidemiology, and End Results (SEER) database). All studies except three [4,21,31] utilized retrospectively collected data for their model developments. The study by Kiritan et al. incorporated both retrospectively and prospectively collected data for the rapid diagnosis of colorectal liver metastasis [31].

A wide range of features were used for the model development in the included studies. A common set of features has radiomics features such as CT, PET/CT, and MRI scans [1,17,18,25,33,49] extracted from medical imaging. These radiomics features encompassed quantitative characteristics of tumor appearance and phenotype from the images. Additionally, some studies extracted semantic features [25], and other quantitative descriptors such as texture and intensity features, from medical images [23,46]. Many studies incorporated clinical features such as patient demographics (e.g., age, gender, and race), tumor characteristics (e.g., size, location, and stage), and biomarkers like Cancer Embryonic Antigen (CEA) and Carbohydrate Antigen (CA19-9) levels [5,11,24,29,30,32,33,37,38,42,43]. Histopathological features from tissue samples, including tumor grade, lymph node metastasis status, and other morphological characteristics, were utilized in several studies [8,16,20,48].

### 3.4. Clinical Focus of the Developed Models

The clinical application of the developed models primarily focused on diagnosing or predicting metastasis to various anatomical sites. Specifically, 25 studies focused on metastasis to lymph nodes, 13 studies on metastasis to the liver, 4 studies on metastasis to the peritoneum, 2 studies on local invasion, and 1 study did not provide information about the origin of metastasis. Maaref et al.’s study was focused on the differentiation of treated and untreated metastatic liver lesions [34].

Thirty-nine studies aimed to utilize their models for various clinical purposes, including primary diagnosis, treatment planning, pre-resection surgery planning, predicting patient prognosis and survival, and risk stratification after diagnosis. Additionally, four studies aimed to apply models during or after treatment, such as evaluating response to therapy [18,27,40,42]. Furthermore, two studies developed models specifically for intraoperative decision-making [21,44].

### 3.5. Data Preparation

In the included studies, 28 articles did not employ any appropriate preprocessing techniques for handling missing values. Six papers utilized various traditional autoscaling and normalization methods. In two of the studies, conventional methods for imputing missing values, such as multivariate imputation based on Chained Equations (MICE), were used in conjunction with a mix of under- and oversampling. This was implemented in the Random Oversampling Examples (ROSE) strategy from the “ROSE” R package [38] and Random Forest (RF) imputation [38]. Three additional papers [4,11,28] employed various sampling preprocessing methods, such as the Synthetic Minority Oversampling Technique (SMOTE) algorithm [4,11,44] and bicubic resampling.

### 3.6. AI Models

The most applied algorithms include support vector machines (SVMs), Random Forests (RFs), and DL models like Convolutional Neural Networks (CNNs) and deep neural networks (DNNs). These models are primarily used for classification tasks, with some studies employing them for regression or a combination of both. Several studies also used advanced ensemble methods and hybrid models, combining different algorithms to enhance performance. Studies applied specialized techniques like Lasso models, Gradient Boosting Machines (GBMs), and radiomic models to increase performance. Overall, the diversity of AI algorithms reflects the broad range of approaches researchers are exploring to tackle the challenges in the domain of metastatic gastrointestinal cancer. Table 7 presents descriptions of various models developed in the included studies.

### 3.7. Validation and Evaluation

Among the 46 articles reviewed, 16 employed K-fold cross-validation to evaluate model performance. Of these, eight studies used 10-fold cross-validation [15,19,26,27,31,36,41,44], and the remaining eight applied 5-fold cross-validation [3,10,11,16,25,37,42,49]. Additionally, one study implemented a 100 × random-split cross-validation method for validation [6]. When considering the reliability and robustness of AI models, 10-fold cross-validation is generally regarded as more thorough than 5-fold cross-validation. This is because 10-fold cross-validation involves more iterations, which helps minimize variance and offers a more consistent assessment of model performance. As such, studies using 10-fold cross-validation are often seen as providing more robust evidence supporting the validity of their findings. In contrast, 5-fold cross-validation, while less computationally demanding, remains a common and practical method for model validation, especially when computational resources are limited. The random-split cross-validation approach, as seen in [6], is less frequently used and may be subject to greater variability due to the random division of data, which can introduce biases depending on how the data are split. However, despite its limitations, it can still offer valuable insights, particularly when more computationally intensive techniques are not feasible.

In summary, while 10-fold cross-validation is typically favored for its greater reliability and validity, each of these techniques plays a valuable role in model evaluation. The choice of method ultimately depends on the specific goals of the study, the design, and the available computational resources.

Table 8 shows explanations of the standard evaluation metrics of AI models considered in studies. The area under the curve (AUC) metric and other metrics such as ROC [60], Sensitivity, Precision, Specificity, and Accuracy were used in 37 articles. In six of these, AUC was the sole evaluation metric. Accuracy was another prevalent evaluation metric, and was deployed for performance assessment in 30 articles. Eight of the articles presented different types of evaluation metrics such as C-index and confusion matrix [1,5,18,19,27,30,42,45], minimal depth, variable importance [29], intersection over union (IoU) mean and Standard deviation (StDev) plus SBD mean and StDev [16], Kaplan–Meier survival curves [2], F1-score, and Precision [18,36,38,44,49]. Less common metrics used in the mentioned studies included combinations of evaluation metrics such as AUROC, *p*-value, NRI/IDI, DCA [26], and Kaplan Meier survival curves [2].

### 3.8. Risk of Bias Assessment

The results of ROB using PROBAST are provided in Table 9. Based on the PROBAST assessments, the domain with the highest risk of bias was “analysis”, mainly due to inadequate handling of confounders, insufficient external validation, and unclear methods for addressing missing data. Only two papers had an ROB across all domains. In contrast, “predictors” and “outcomes” consistently showed a low risk of bias across all papers, as they were well defined and measured using standardized techniques and reliable methods. Most studies had high risk of bias in the “participants” domain due to small sample sets, selection bias, and limited generalizability from single-center designs.

## 4. Discussion

Our systematic literature review highlights the significant advancements and applications of AI in the diagnosis and treatment of metastatic gastrointestinal cancers. The results have shown promise in improving the accuracy and efficiency of cancer metastasis detection, treatment planning and predicting its outcomes, and identifying prognostic biomarkers. In recent years, DL algorithms, such as CNNs, U-Net, and ResNet, have been increasingly used to construct predictive models, demonstrating higher diagnostic accuracy compared to traditional radiologist evaluations [45,49]. Additionally, ML-based radiomics models have been effective in predicting local tumor progression and the occurrence of metachronous liver metastases in colorectal cancer patients [1,3].

The integration of AI into the clinical management of gastrointestinal cancers holds significant promise for enhancing diagnostic accuracy, personalizing treatment plans, and improving patient outcomes. AI models, such as DL algorithms, can detect subtle patterns in imaging data indicative of metastasis or disease progression—patterns often missed by traditional methods [49]. This capability enables precise risk stratification and predictive analytics, supporting informed clinical decision-making and timely therapeutic interventions [1,2,46]. Additionally, AI can streamline clinical workflows by automating routine tasks, allowing clinicians to focus on complex patient care issues [4]. As these models undergo further validation, their adoption could transform the diagnosis and treatment of gastrointestinal cancers, ultimately improving survival rates and enhancing patients’ quality of life [3,45]. In this regard, one significant opportunity lies in AI’s potential to support real-time decision-making by integrating predictive models with EHRs and imaging systems. For instance, AI-driven tools can analyze patient data to predict complications, recommend treatment plans, or identify candidates for personalized therapies. Such systems could enable earlier interventions and reduce morbidity and mortality [62].

Despite these promising opportunities, the development, implementation, and application of AI in clinical settings also face several barriers and challenges. In the development phase, for example, the collection and preparation of datasets are crucial in applying AI techniques, as the performance of AI models largely depends on the quality and size of the data [62]. AI models like RF and SVM are among the most commonly used algorithms due to their robustness in handling large, high-dimensional datasets and their ability to extract relevant features for classification tasks [5,7,45]. For instance, RF has been employed to build predictive models based on clinical and radiomics features, while SVM has been used for binary classification tasks, such as distinguishing cancerous tissues from non-cancerous tissues [6,8,10]. However, data preprocessing remains a significant challenge, as clinical datasets often require transformation, cleaning, and handling of missing values before they can be effectively used. This preprocessing stage can significantly influence the final model, making it crucial to elucidate the specific steps involved [63]. To address these challenges proactively, strategies such as adopting interoperable standards, establishing centralized data management teams, and implementing automated preprocessing pipelines can streamline these processes. While standardizing data formats, addressing missing values, and ensuring consistent labeling across institutions are resource-intensive tasks, they are essential for enhancing the reliability and clinical applicability of AI models.

Another major challenge in the development phase is the reliance on retrospective data, which can introduce biases and limit the generalizability of AI models to new patient populations [1,2,46]. Additionally, the variability in imaging protocols and the quality of clinical data across studies complicates the standardization of AI applications. Small sample sizes in many studies reduce statistical power and reliability, making it difficult to draw definitive conclusions about the efficacy of AI algorithms [6,9]. Other technical challenges, besides the preprocessing of data (see above), are segmentation, feature extraction, and the computational demands of training DL models [7,8]. Moreover, the need for robust validation of AI models in real clinical settings and their interpretability in high-stakes clinical scenarios remain critical issues for care providers to trust these models’ recommendations. While these models can achieve high levels of accuracy, understanding the underlying decision-making process is often difficult, which can impede clinical trust and acceptance [8,10,45]. Clinicians may be reluctant to rely on black-box algorithms without clear explanations of how decisions are made. Ongoing research to refine algorithms for clinical utility, develop interpretable AI models, incorporate XAI techniques, and provide intuitive visualizations, alongside educational initiatives to improve clinicians’ understanding of AI and its capabilities, can help address these concerns and foster trust among healthcare providers [64].

To overcome the existing challenges, future research should prioritize the use of prospective data to enhance the validity and applicability of AI models in real-world clinical settings. Standardizing preprocessing techniques and ensuring comprehensive risk of bias assessments will improve the reliability of studies [49]. Increasing sample sizes and diversifying datasets will help in developing more robust and generalizable models [9,45]. There should be a concerted effort to establish common methodological standards and guidelines, facilitating better comparison and synthesis of research findings across different studies. Collaboration between multidisciplinary teams, including clinicians, data scientists, and engineers, is crucial to advancing the field and achieving meaningful clinical integration of AI technologies [1,2]. Moreover, improving the interpretability of AI models is critical for building clinical trust and acceptance [8,10,45]. Developing explainable AI techniques that provide insights into the decision-making process can help bridge the gap between AI predictions and clinical decision-making [8,10,11].

In conclusion, while AI holds great promise for enhancing the diagnosis and treatment of gastrointestinal cancers, addressing its clinical and technical challenges is essential to realize its potential fully. As these models are further validated and refined, AI could lead to a paradigm shift in the management of gastrointestinal cancers, ultimately improving patient outcomes and quality of life [3,45].

### 4.1. Challenges and Recommendations for Future Research

Many of the studies reviewed in this analysis exhibit strengths in their application of AI to metastatic gastrointestinal cancers. Notably, numerous studies utilized sophisticated machine learning methods such as RF, SVM, and DNNs, which are particularly effective for managing complex, high-dimensional datasets. These approaches were chosen for their ability to uncover significant patterns in data and improve predictive accuracy. In addition, several studies incorporated internal validation techniques like cross-validation to assess the reliability of the models. The use of diverse clinical features, including radiomic data, histopathological information, and patient demographics, highlights a comprehensive feature selection process, which is essential for improving the models’ predictive performance and generalizability across different populations.

However, several limitations in the reviewed studies reduce their broader applicability and impact. One key challenge is the predominance of retrospective data collection. Data in healthcare are very contextualized [65], which introduces potential biases and undermines the reliability of the findings. The absence of prospective data and real-time validation further limits the ability to evaluate how these AI models would perform in actual clinical environments. Additionally, many studies did not apply appropriate preprocessing methods to address issues such as missing data or noise, which can adversely affect the models’ robustness and accuracy. Another notable limitation is the small sample size of many studies, often involving fewer than 200 patients, which restricts the external validity of their conclusions. Furthermore, inconsistencies in reporting essential methodological details—such as data preprocessing steps and bias risk assessments—make it difficult to assess the quality and reproducibility of these studies.

Challenges in data preprocessing represent a significant bottleneck in the successful implementation of AI models in clinical practice. Clinical datasets are often characterized by heterogeneity, missing values, and imbalances, which can compromise the performance and reliability of predictive models. Effective preprocessing strategies, such as data cleaning, normalization, and imputation, are essential for ensuring data quality and consistency. Techniques like SMOTE can be used to address class imbalances, while advanced imputation methods, such as multiple imputation or matrix factorization, can handle missing data effectively. Furthermore, adopting automated preprocessing pipelines, leveraging tools like AutoML, could reduce human effort and variability in preprocessing steps, enhancing reproducibility and efficiency.

Model interpretability remains another critical challenge in translating AI advancements into clinical applications. DL algorithms are often criticized for their lack of transparency, which can impede clinical adoption. To address this, XAI methods should be integrated into the development and validation processes of AI models. Techniques such as SHAP (Shapley Additive Explanations) and LIME (Local Interpretable Model-agnostic Explanations) can provide insights into how models derive their predictions, helping clinicians understand and trust AI-based recommendations.

In addition to technical solutions, clear communication of AI results to end-users, such as clinicians and patients, is paramount. Interactive dashboards and visualization tools that translate complex AI outputs into easily interpretable formats could significantly enhance usability. Moreover, involving clinicians in the development and validation of AI models can ensure that interpretability efforts align with real-world clinical needs. To improve the reliability and generalizability of future research in this area, several recommendations can be made. First, prospective studies that include real-time validation should be prioritized to better assess the clinical effectiveness of AI models in practice. Second, adopting standardized preprocessing methods is crucial to ensure that data issues, such as missing values and noise, do not undermine model performance. Lastly, future studies should aim to work with larger, more diverse datasets to enhance the external validity of the findings and mitigate the biases inherent in small sample sizes. By implementing these strategies, future research can improve the clinical applicability and reproducibility of XAI methods in metastatic gastrointestinal cancer.

### 4.2. Risk of Bias Considerations

While our review presents several notable strengths, it is important to acknowledge certain limitations. A primary concern is the potential for publication bias, which is a common issue in many systematic reviews. Studies that demonstrate positive or high-performance results are more likely to be published, whereas studies with negative or inconclusive findings may be underreported. This bias could lead to an overestimation of the effectiveness of AI in the context of metastatic gastrointestinal cancer. Furthermore, the review is based on studies published within a defined time frame. Given the rapid advancements in AI technology, this temporal scope may limit the review’s ability to fully capture the most recent developments in the field. As AI continues to progress quickly, reviews such as this one may become outdated over time, making it difficult to incorporate the latest innovations. Nonetheless, reviews like the one presented here are crucial for identifying existing gaps in the literature and guiding future research priorities.

Another key limitation found across the reviewed studies—and one that requires further attention—is the lack of real-world validation of AI models within clinical settings. While many AI tools have been tested in controlled environments, their performance and applicability in the dynamic and varied conditions of clinical practice remain insufficiently explored. As AI technologies continue to evolve, future studies should prioritize large-scale, prospective trials that assess the effectiveness of AI in real-world clinical settings. Such studies are critical for understanding how AI can be effectively integrated into everyday clinical practice and contribute to improving patient outcomes.

An important methodological limitation identified during our systematic review is the variability in the reporting of study quality and risk of bias, as evaluated using PROBAST. The PROBAST assessment revealed that many studies exhibit high or unclear risks of bias in key domains, including participant selection, outcome definition, and statistical analysis. These shortcomings can significantly influence the reliability and generalizability of AI models in clinical applications.

One major challenge identified in the PROBAST assessment is the inappropriate handling of participant selection, often due to retrospective study designs and limited sample diversity. Many studies included homogeneous patient populations, which may fail to represent the demographic and clinical heterogeneity of real-world settings. Future research should prioritize the inclusion of diverse patient cohorts and employ prospective study designs to minimize selection bias and enhance model applicability across different populations.

Another area of concern is the lack of standardized outcome definitions, as inconsistent criteria for defining endpoints such as metastasis or treatment response can lead to discrepancies in model performance across studies. Establishing universally accepted definitions and reporting guidelines, such as those proposed by the TRIPOD (Transparent Reporting of a Multivariable Prediction Model for Individual Prognosis or Diagnosis) initiative, is crucial for improving consistency and facilitating the comparison of results.

To address these limitations, future studies should integrate the PROBAST framework into their design and reporting processes, ensuring a more rigorous assessment of study quality and risk of bias. By addressing these challenges, the field can move toward producing more reliable and clinically translatable AI models and more state-of-the-art methods, ultimately improving patient care and outcomes in metastatic gastrointestinal cancer management.

## 5. Conclusions

This systematic literature review demonstrates the potential of AI in enhancing the diagnosis and treatment of metastatic gastrointestinal cancers. The domain’s status reflects promising advancements but underscores the need for more rigorous and standardized research methodologies. Future research should focus on prospective data collection, standardizing preprocessing techniques, and conducting comprehensive risk of bias assessments. Increasing sample sizes and diversifying datasets coming from different care-providing contexts will enhance model robustness and generalizability. Furthermore, giving priority to and improving the interpretability of AI models, especially for clinical users of such AI models, will build clinical trust and lead to better patient outcomes. Overall, while AI shows significant promise, addressing these methodological challenges is crucial for its successful integration into clinical practice.

## Figures and Tables

**Figure 1 cancers-17-00558-f001:**
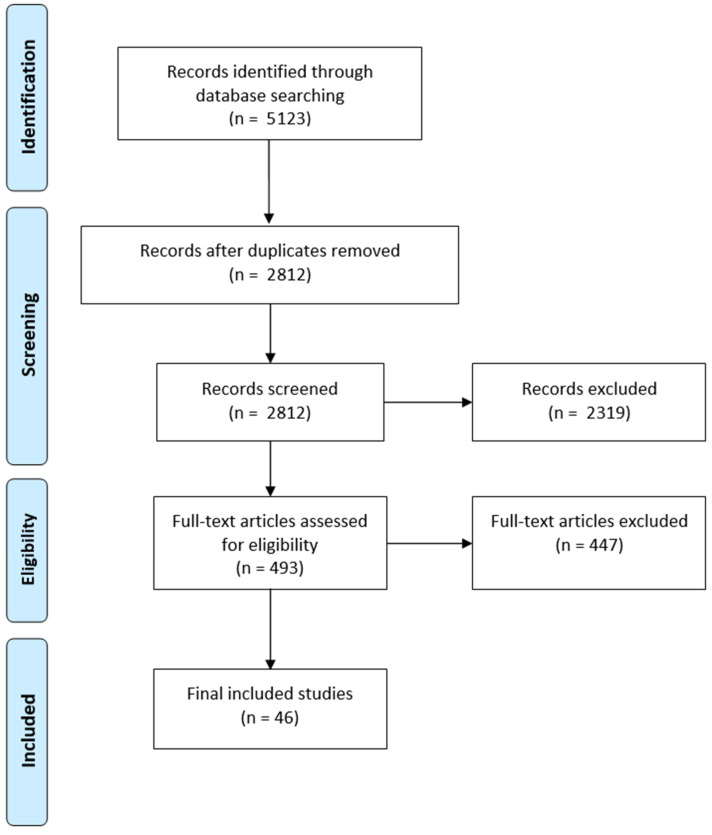
Flow diagram of study selection (PRISMA chart).

**Table 1 cancers-17-00558-t001:** Research questions.

Q1	What AI techniques have been used in different applications of metastatic gastrointestinal cancers?
Q2	What are the common clinical features used in studies?
Q3	What are the common preprocessing steps for AI models’ development?
Q4	What are the methodology settings?
Q5	What are the challenges and research gaps in this domain?

**Table 2 cancers-17-00558-t002:** Search criteria.

G1—AI keywords	Artificial Intelligence, machine learning, learning algorithms, deep learning, unsupervised machine learning, supervised learning
G2—Medical keywords	Gastrointestinal Neoplasms, Digestive System Neoplasms, Esophageal Neoplasms, Stomach Neoplasms, Colorectal Neoplasms, Liver Neoplasms, Rectal Neoplasms, Biliary Tract Neoplasms, Pancreatic Neoplasms, Peritoneal Neoplasms, Cancer, Metastasis, Neoplasm Metastasis
G3—Document type	Journal
G4—Publication year	1 January 2010–1 January 2022
G5—Final result	G1 AND G2 AND G3 AND G4

**Table 3 cancers-17-00558-t003:** Inclusion and exclusion criteria.

Inclusion Criteria	Exclusion Criteria
Cohort should be metastasis gastrointestinal cancer patients.	Studies with traditional statistical models.
Developing AI techniques for metastatic gastrointestinal cancer should be the main aim.	Not journal articles.
Articles should be journal publications in English.	Not English language publications.
	Studies with metastatic gastrointestinal cancer patients as a subgroup of the cohort.
	The main focus of the study is not an explicit application for metastatic gastrointestinal cancer patients.

**Table 4 cancers-17-00558-t004:** Studies’ sample numbers.

Number of Patients	Number of Studies
<100	9
101–200	13
201–300	3
301–400	5
401–500	1
501–600	1
601–700	1
701–800	2
1001–2000	4
2001–4000	3
>4001	4

**Table 5 cancers-17-00558-t005:** Characteristics of included studies.

ID	Authors	Year	Country	Study Type	Outcome		Age (Year) Mean ± Std or Median	Male (%)
					Population	Patient Type		
A1	An et al. [2]	2022	China	Retrospective	148	PDA	59.20 ± 10.60	49.32
A2	Tomita et al. [17]	2021	Japan	Retrospective	201 lymph nodes in 23 OSCC cases	OSCC	52.00 ± 8.00	56.52
A3	Feng et al. [15]	2019	China	Retrospective	490	GC	61.80 ± 10.40	74.08
A4	Taghavi et al. [18]	2020	The Netherlands	Retrospective	91	CC	64.00 ± 11.00	60.43
A5	Huang et al. [19]	2020	China	Retrospective	544	AGC	60.0 (median)	65.99
A6	Dong et al. [1]	2020	China, Italy	Retrospective	730	LAGC	74.22 ± 13.95	64.52
A7	Mermod et al. [20]	2020	Switzerland, Australia	Retrospective	168	Early-stage OSCC	62.09	62.50
A8	Schnelldorfe et al. [21]	2019	USA	Prospective	35	GC	67.00	65.71
A9	Liu et al. [22]	2021	China	Retrospective	185	GC	62.00	68.60
A10	Takeda et al. [23]	2017	Japan	Retrospective	242	Non-neoplasms, adenomas, and invasive cancers	64.75 ± 11.45	61.98
A11	Kasai et al. [24]	2021	Japan	Prospective	323	Primary RC surgery with LLND	65.00	NA
A12	Wang et al. [5]	2021	China	Retrospective	159	T1-2 GC	61.78 ± 10.47	71.06
A13	Zhang et al. [16]	2011	China	Retrospective	175	GC	59.80	71.42
A14	Shi et al. [25]	2020	China	Retrospective	159	CRLM	NA	61.00
A15	Chuang et al. [8]	2021	Taiwan	Retrospective	1051	CC	NA	NA
A16	Kang et al. [26]	2021	Republic of Korea	Retrospective	316	AGC and T1 CRC	NA	57.59
A17	Mühlberg et al. [27]	2021	Germany	Retrospective	103	CRLM	61.00 ± 11.20	53.40
A18	Mirniaharikandehei et al. [28]	2021	USA	Retrospective	159	With and without PM	Cases with PM = 59.49 ± 11.97 Cases without PM = 59.11 ± 8.75	Cases with PM = 59.10 Cases without PM = 18.80
A19	Rice et al. [29]	2017	USA, China, Finland, Canada, Spain	Retrospective	5806	Esophagectomy alone	63.00 ± 11.00	77.00
A20	Chen et al. [30]	2019	China	Retrospective	146	AGC	64.94 ± 11.11	Center 1: 76.05 Center: 80.85
A21	Cancian et al. [7]	2021	Italy, UK	Retrospective	303	CLRM	NA	NA
A22	Yang et al. [4]	2019	China	Retrospective	100	ECC	57.10 ± 10.00	54.00
A23	Kiritani et al. [31]	2021	Japan	Retrospective and prospective	183	With and without CRLM	68.00	59.20
A24	Chen et al. [32]	2020	China	Retrospective	733	Superficial esophageal squamous cell carcinoma (SESCC)	62.80	69.98
A25	Liu et al. [33]	2021	China	Retrospective	186	Rectal adenocarcinoma	59.22 ± 5.72	68.81
A26	Maaref et al. [34]	2020	Canada	Retrospective	202	CRLM	NA	NA
A27	Zhou et al. [35]	2020	China	Retrospective	1080	GC with CT	63.7 ± 11.65	77.68
A28	Starmans et al. [6]	2021	The Netherlands, Belgium	Retrospective	76	Pure HGPs	68.00	57.89
A29	Zhou et al. [36]	2021	China	Retrospective	30	Metastatic solid tumors	NA	NA
A30	Bur et al. [37]	2019	USA	Retrospective	2032	Clinically node negative OCSCC	NCDB: 61.90, single institution: 58.10	NA
A31	Ahn et al. [38]	2021	Republic of Korea	Retrospective	26,733	Early CRC (T1)	NA	52.81
A32	Ariji et al. [39]	2020	Japan	Retrospective	51	CLNM from OCSCC	64.00	52.94
A33	Ariji et al. [10]	2018	Japan	Retrospective	45	OCSCC	63.00	53.33
A34	Dercle et al. [40]	2020	USA, France, Belgium, Germany,	Retrospective	667	Liver metastatic CRC	NA	64.72
A35	Kwak et al. [41]	2021	Republic of Korea	Retrospective	16,878	Gastric metastasis	metastasis: 64.60 ± 14.40, non-metastasis: 62.60 ± 13.70	metastasis: 58.60, non-metastasis: 61.40
A36	Taghavi et al. [42]	2021	The Netherlands, Spain	Retrospective	90	Colorectal liver metastases	62.00 ± 11.00	57.77
A37	Gupta et al. [43]	2019	Taiwan	Retrospective	4021	CRC	NA	56.93
A38	Zhong et al. [9]	2021	China, Germany	Retrospective	313	SCC	55.07 ± 12.46	60.38
A39	Sitnik et al. [44]	2021	Croatia	Retrospective	19	Metastatic colon cancer	NA	NA
A40	Li et al. [45]	2021	China	Retrospective	129	Rectal cancer	58.40 ± 10.27	64.34
A41	Li et al. [3]	2019	China, USA	Retrospective	48	Liver metastasis (LM) in colon cancer (CC)	61.52 ± 12.53	62.50
A42	Li et al. [46]	2020	China	Retrospective	3364	CRC	NA	NA
A43	Shuwen et al. [47]	2020	China	Retrospective	1186	CAD	NA	NA
A44	Takamatsu et al. [48]	2019	Japan	Retrospective	397	CRC	Training: 61.30 ± 11.45 Test: NA	Training: 51.51
A45	Mao et al. [49]	2021	China	Retrospective	114	Metastatic liver cancer	59.10	61.40%
A46	Lee et al. [11]	2020	Republic of Korea, USA	Retrospective	2019	CRC	62.70 ± 9.35	62.85

Adenocarcinoma; CC, colorectal cancer; CLNM, cervical lymph node metastases; CRLM, colorectal liver metastasis; CT, computer tomography; EC, external cohort; ESC, esophageal squamous carcinoma; FNB, fine-needle biopsy; GC, gastric cancer; LM, liver metastasis; CRC, colorectal cancer; LAGC, locally advanced gastric cancer; IC, internal cohort; LC, liver cancer; LLND, lateral lymph node dissection; NA, not available; NM, not mentioned; OCSCC, oral cavity squamous cell carcinoma, including patients with cancer of the tongue, gingivae, and floor of the mouth who underwent cervical node dissection; PDA, pancreatic ductal adenocarcinoma; PM, peritoneal metastasis, RC, rectal cancer; SCC, squamous cell carcinoma.

**Table 6 cancers-17-00558-t006:** Selected articles and their AI-related characteristics.

Id	AI Algorithm	Evaluation Metrics	Handling Missing Values	Hyperparameter Optimization	Approach	Validation
A1	LR, SVM, Resnet 18	AUC = 0.92, Confidence, Accuracy = 0.86, Sensitivity = 0.92, Specificity = 0.78, PPV = 0.8, NPV = 0.93	No	NA	Classification	Internal
A2	SVM	*p*-value = 0.05, AUC = (0.820 at level I/II, 0.820 at level I, and 0.930 at level II), Cutoff, Accuracy, Sensitivity, Specificity	No	SVM-RBF	Classification	Internal
A3	SVM	AUC = (0.699–0.833), Sensitivity, Specificity, PPV, NPV, Accuracy = 71.3%, Cutoff Value	No	10-fold cross-validation, Monte Carlo cross-validation (200 repeats), RBF	Classification	Internal
A4	RF	AUC = 71% and 86%, F1-score, CI = (69–72%, 85–87%)	No	Bayesian hyperparameter optimization	Classification	Internal
A5	DCNN	Sensitivity = 81%, AUC = 0.670, 95% CI: 0.615–0.739; *p* < 0.001, Specificity = 87.5%	No	NA	Forecasting	Internal
A6	DLRN	C-index = 0.821, confusion matrix	No	NA	Forecasting	Internal and external
A7	RF, Lasso LR, SVM, C5.0	Sensitivity, Specificity, NPV, PPV, Accuracy = 0.88, AUC = 0.89	No	NA	Classification	External
A8	DNN	Sensitivity, Specificity, PPV, NPV, Accuracy, AUC = 0.47	No	NA	Classification	Internal
A9	Balanced Bagging Ensemble Classifier	Accuracy = 0.852, AUC = 0.822, Sensitivity = 0.733, Specificity = 0.891, PPV = 0.688, NPV = 0.911	No	NA	Classification	Internal
A10	SVM	Sensitivity = 89.4%, Specificity = 98.9%, Accuracy = 94.1%, PPV = 98.8%, NPV = 90.1%	No	NA	Classification	Internal
A11	Prediction One (Sony Network Communications) Software	AUC = 0.754, U-test = 0.022, Accuracy = 80.4%, Sensitivity = 90.0%, Specificity = 79.4%, PPV, NPV, *p*-value = 0.022	No	NA	Classification	Internal
A12	LR	Confusion matrix, Accuracy = 0.899, Sensitivity = 0.882, Specificity = 0.903, PPV = 0.714, NPV = 0.966, AUC = 0.908, *p*-value	No	NA	Forecasting	Internal
A13	SVM	T-test, U-test, AUC, AUC = 0.876, Sensitivity = 88.5%, Specificity = 78.5%, *p*-value = 0.002, *p*-value < 0.001	No	5-fold cross-validation	Classification	Internal
A14	ANN, KNN, SVM, Bayes, LR, AdaBoost, GB	AUC = 0.95, Accuracy = 87.10%, Sensitivity = 89.19%, Specificity = 84.00%, PPV = 89.19%, NPV = 84.00%, *p*-value	No	NA	Classification	Internal
A15	ResNet-50	AUC = 0.9724, Accuracy = 98.50%	No	NA	Classification	Internal
A16	Lasso regression	AUROC (0.765 vs. 0.518, *p* = 0.003), NRI (0.447, *p* = 0.039)/IDI (0.121, *p* = 0.034), DCA, *p*-value	No	log (λ), where λ is a tuning hyperparameter, cross-validation	Classification	Internal
A17	LR, RF	AUC = 0.70, Z-value, *p*-value, Odds ratio, C-index = [0.56, 0.90]	No	10-fold cross-validation	Classification	Internal
A18	DT, RF, SVM, LR, GBM	Precision = 65.78%, Sensitivity = 43.10%, Specificity = 87.12%, Accuracy = 71.2%, AUC = 0.69 ± 0.019	SMOTE	Cross-validation	Classification	Internal
A19	RF	Minimal depth, variable importance, probability	No	NA	Classification	Internal
A20	LASSO, LR, and Learning Vector Quantization (LVQ)	U-test, *p*-value, AUC = 0.657, Accuracy = 0.745, Sensitivity = 0.853, Specificity = 0.462, confusion matrix	No	Cross-validation	Classification and Regression	Internal and external
A21	UNet, SegNet, DeepLab-v3	IoU mean = 89.13, IoU StDev = 3.85, SBD mean = 79.00, SBD StDev = 3.72	No	NA	Classification	Internal
A22	RF	AUC = 0.80 and 0.90	SMOTE	NA	Classification	Internal
A23	LR	Specificity = 100%, Sensitivity = 99%, Accuracy = 99.5%, *p*-value	No	10-fold cross-validation	Classification	Internal
A24	LR, ANN	Specificity = 91.20%, Sensitivity = 87.06%, Accuracy = 90.72%, *p*-value, AUC = 0.915, PPV = 56.49%, NPV = 98.17%, NRI = −1.1%, IDI = 23.3%	No	Cross-validation	Classification	Internal
A25	SVM	AUC = 0.827, *p*-value, Sensitivity = 0.815, Specificity = 0.694, PPV = 0.667, NPV = 0.833	No	NA	Classification	Internal
A26	DCNN	AUC, Sensitivity, Specificity, Accuracy = 91%,78%	No	NA	Classification	Internal
A27	Light Gradient Boosting Machine, GradientBoosting, RF, Logistic, and DT	AUC = 0.745, Accuracy = 0.907, MSE = 0.093	No	Tuning parameter for each model	Classification	Internal
A28	CNN, LR, SVM, RF, Quadratic Discriminant Analysis, AdaBoost, Extreme Gradient Boosting	AUC = 0.72, Accuracy = 0.65, Sensitivity = 0.62, Specificity = 0.68	No	Cross-validation	Classification	Internal
A29	RFC and SVM	AUC = 0.999, Sensitivity = 1, Specificity = 0.997, Accuracy = 0.998, PPV = 0.954, F1 = 0.977	No	Cross-validation	Classification	Internal
A30	LR, RF, SVM, GB	Specificity, Sensitivity, AUC = 0.840, *p*-value	No	Cross-validation	Classification	Internal and external
A31	LR, XGB, KNN, CARTs, SVM, NN, RF	AUC = 0.991, Accuracy = 0.960, Sensitivity = 0.997, Specificity = 0.929, Precision (PPV) = 0.919, NPV = 0.998, FDR = 0.081, AP = 0.995, F1-score = 0.956, MCC = 0.922	Random oversampling	Cross-validation	Classification	Internal
A32	AlexNet	AUC, Accuracy = 84.0%, Sensitivity, Specificity, PPV, NPV	No	NA	Classification	Internal
A33	CNN	Accuracy = 78.2%, Sensitivity = 75.4%, Specificity = 81.0%, PPV = 79.9%, NPV = 77.1%, AUC = 0.80	No	Cross-validation	Classification	Internal and external
A34	Deep learning, RF	AUC = 0.80	No	NA	Classification	Internal
A35	CART, KNN, LR, RF, SVM, XGB	AUC = 0.956, Sensitivity, Specificity, AP, F1-score, MCC	No	Cross-validation	Classification	Internal
A36	Three machine learning survival models	C-index = 0.77–0.79, *p*-value	No	Bayesian hyperparameter optimization, cross-validation	Classification	Internal and external
A37	RF, SVM, LR, MLP, KNN, AdaBoost	Accuracy = 0.89, Precision = 0.89, Recall = 0.88, F-measure = 0.89, AUC = 0.94	No	Scikit-Optimize, Cross-validation	Classification	Internal
A38	ANN	Accuracy = 84.1%, Sensitivity = 93.1%, Specificity = 76.5%, AUC = 0.943, net reclassification index (NRI) = 40%	No	NA	Classification	Internal
A39	SVM, KNN, U-Net, U-Net++, DeepLabv3	F1-score = 83.67%, Accuracy = 89.34%, TPR, TNR, BACC, PPV = 81.11%	No	Cross-validation	Classification	Internal
A40	Inception-v3	Accuracy = 95.7%, PPV = 95.2%, NPV = 95.3%, Sensitivity = 95.3%, Specificity = 95.2%, AUC = 0.994, confusion matrix, *p*-value > 0.05	No	NA	Classification	Internal
A41	SVM	Accuracy = 69.50%, Specificity = 83.14%, Sensitivity = 62.00%, area under the curve (AUC) = 0.69	No	Cross-validation	Classification	Internal
A42	AB, MLP, LeNet, DT, NB, AlexNet, KNN, SGD, AlexNet Pre-trained, LR, SVM	Accuracy = 0.7583, AUC = 0.7941, Sensitivity = 0.8004, Specificity = 0.7997, PPV = 0.7992, NPV = 0.8009	No	freezing and fine-tuning parameters of CNN models	Classification	Internal
A43	LR, NN, SVM, RF, GBDT, Catboost	Accuracy = 1, AUC = 1	No	Cross-validation	Classification	Internal and external
A44	RF, LR	AUC = 0.94	No	Cross-validation	Classification	Internal
A45	KNN, SVM, RF, LR, MLP	AUC = 0.816 ± 0.088, Accuracy = 0.843 ± 0.078, F1-score, Specificity = 0.880 ± 0.117, Sensitivity = 0.768 ± 0.232, Precision	No	Cross-validation	Classification	Internal
A46	VGG16, Logistic Regression, Random Forest	AUC = 0.747 ± 0.036	SMOTE	Cross-validation	Classification	Internal

ANN, artificial neural network; AUC, area under the curve; CART, classification and regression tree; DNN, deep neural network; DT, decision tree; FPR, false positive rate; GBM, Gradient Boosting Machine; IOU, intersection over union; KNN, K-nearest neighbors; LASSO, Least Absolute Shrinkage and Selection Operator; LR, logistic regression; MCC, Matthews correlation coefficient; ML, machine learning; MLP: Multi-Layer Perceptron; NLR, negative likelihood ratio; NN, neural network; NPV, negative predictive value; LR: logistic regression, PCA, principle component analysis; PLR, positive likelihood ratio; PPV, positive predictive value; RBF, radial-basis function; RF, Random Forest; ROSE, Random Oversampling Example; SVM, support vector machine.

**Table 7 cancers-17-00558-t007:** Summary of AI algorithms.

Algorithm	Description	Pros	Cons
LR [50]	LR is a supervised ML algorithm adopted from linear regression. It can be used for classification problems and finding the probability of an event happening.	Fast training, good for small datasets, and easy to understand.	Not very accurate, not proper for non-linear problems, high chance of overfitting, and not flexible enough to adopt to complex datasets.
DT [51]	DT is a supervised ML algorithm that solves a problem by transforming the data into a tree representation where each internal node represents an attribute and each leaf denotes a class label. CART is also a decision tree algorithm used for both classification and regression tasks. It is a supervised learning algorithm that learns from labeled data to predict unseen data.	Easy to understand and interpret, robust to outliers, no standardization or normalization required, and useful for regression and classification.	High risk of overfitting, not suitable for large datasets, and adding new samples leads to the regeneration of the whole tree.
CART [52]	CART is used for classification and regression by splitting the data into subsets to achieve the highest information gain or lowest variance.	Easy to understand and interpret, robust to outliers, no standardization or normalization required, and useful for regression and classification.	High risk of overfitting, not suitable for small datasets; adding new samples leads to the regeneration of the whole model.
KNN [53]	KNN is a supervised and instance-based ML algorithm. It can be used when we want to forecast a label of a new sample based on similar samples with known labels. Different similarity or distance measures such as Euclidean can be used.	Simple and easy to understand, easy to implement, no need for training, and useful for regression and classification.	Memory-intensive, costly, slow performance, and all training data might be involved in decision-making.
SVM [54]	SVM is an instance-based and supervised ML technique that generates a boundary between classes known as a hyperplane. Maximizing the margin between classes is the main goal of this technique.	Efficient in high-dimensional spaces. Effective when the number of dimensions exceeds the number of samples, useful for regression and classification, regularization capabilities that prevent overfitting, and handling non-linear data.	Not suitable for large datasets, not suitable for noisy datasets, regularization capabilities that prevent overfitting, handling non-linear data, and long training time.
GB [55]	GB is a supervised ML algorithm, which produces a model in the form of an ensemble of weak prediction models, usually DT. GB is an iterative gradient technique that minimizes a loss function by iteratively selecting a function that points toward the negative gradient.	High accuracy, high flexibility, fast execution, and useful for regression and classification, robust to missing values and overfitting.	Sensitive to outliers, not suitable for small datasets, and many parameters to optimize.
RF [56]	RF is an ensemble and supervised ML algorithm that is based on the bagging technique, which means that many subsets of data are randomly selected with replacements and each model such as DT is trained using one subset. The output is the average of all predictions of various single models.	High accuracy, fast execution, useful for regression and classification, and robust to missing values and overfitting.	Not suitable for limited datasets; may change considerably by a small change in the data.
ANN [57]	ANN is a family of supervised ML algorithms. It is inspired by the biological neural network of the human brain. ANN consists of input, hidden, and output layers and multiple neurons (nodes) carry data from the input layer to the output layer.	Accurate; suitable for complex non-linear classification and regression problems.	Very slow to train and test, large amounts of essential data, computationally expensive, and prone to overfitting.
DNN [58]	DNN is a family of supervised ML algorithms. DNN is based on NNs where the adjective ’deep’ comes from the use of multiple layers in the network. Usually having two or more hidden layers counts as a DNN. Some specific training algorithms and architectures exist, such as LSTM, GAN, and CNN for DNNs. DNNs provide the opportunity to solve complex problems when the data are very diverse, unstructured, and interconnected.	High accuracy, features are automatically deduced and optimally tuned, robust to noise, and architecture is flexible.	Needs a very large amount of data, computationally expensive, not easy to understand, no standard theory in selecting the right settings, and difficult for less skilled researchers.
Lasso [59]	LASSO is a regularization technique used in statistical modeling and machine learning for estimating the relationships between variables and making predictions.	Simplicity, feature selection, and robustness.	Introduces bias into the estimates; low performance when the number of observations is less than the number of features or there is high multicollinearity among the features.

ANN, artificial neural network; CART: classification and regression tree; CNN, Convolutional Neural Network; DNN, deep neural network; DT, decision tree; GAN, generative adversarial network; GB, gradient boosting; KNN, K-nearest neighbors; LR, logistic regression; LSTM, long-short term memory network; ML, machine learning; RF, Random Forest; SVM, support vector machine; CART, classification and regression tree; XGB, Extreme Gradient Boosting; LASSO, Least Absolute Shrinkage and Selection Operator.

**Table 8 cancers-17-00558-t008:** Summary of AI algorithms’ evaluation metrics.

Metric	Description
Accuracy [61]	Accuracy is a general metric that quantifies the proportion of correctly classified instances (both positive and negative) out of the total instances in the dataset. While Accuracy provides a simple overall performance measure, it may be misleading for imbalanced datasets, as it does not differentiate between the types of errors (false positives vs. false negatives).
Precision [61]	Precision evaluates the proportion of correctly predicted positive cases out of all cases predicted as positive by the model. High Precision indicates that the model is accurate in its positive predictions, minimizing false positives. Precision is especially relevant in scenarios where the cost of false positives is high.
Sensitivity [61]	Sensitivity, also known as the true positive rate, measures the model’s ability to correctly identify positive cases out of all actual positive cases in the dataset. A high Sensitivity value indicates that the model effectively identifies most of the true positive cases, making it particularly important in applications where minimizing false negatives is critical, such as in disease diagnosis.
Specificity [61]	Specificity measures the model’s ability to correctly identify negative cases out of all actual negative cases. A high Specificity value signifies that the model can accurately exclude non-relevant cases, reducing the occurrence of false positives. Specificity is critical in contexts where false positives may lead to unnecessary interventions.
F1-Score [61]	The F1-score is the harmonic mean of Precision and Sensitivity, providing a balanced metric that considers both false positives and false negatives.
AUC (ROC) [61]	The AUC (ROC) is a widely used metric to evaluate the performance of binary classification models by measuring their ability to distinguish between two classes. The ROC curve plots the true positive rate (Sensitivity) against the false positive rate (1-specificity) at various threshold levels. The AUC quantifies the area under this curve, providing a single scalar value ranging from 0 to 1. A higher AUC indicates better model performance, with 1 representing a perfect classifier and 0.5 reflecting no discriminatory power (equivalent to random guessing). The AUC is particularly useful for imbalanced datasets, as it evaluates the model’s performance across different classification thresholds.

**Table 9 cancers-17-00558-t009:** ROB assessment for included studies.

Id	Domain 1: Participants	Domain 2: Predictors	Domain 3: Outcome	Domain 4: Analysis
A1	High	Low	Low	High
A2	High	Low	Low	High
A3	High	Low	Low	High
A4	High	Low	Low	High
A5	Low	Low	Low	Unclear
A6	High	Low	Low	Low
A7	High	Low	Low	High
A8	High	Low	Low	High
A9	High	Low	Low	High
A10	High	Low	Low	High
A11	High	Low	Low	High
A12	High	Low	Low	High
A13	High	Low	Low	High
A14	High	Low	Low	High
A15	Low	Low	Low	High
A16	Unclear	Low	Low	High
A17	High	Low	Low	High
A18	High	Low	Low	High
A19	Low	Low	Low	Unclear
A20	High	Low	Low	Unclear
A21	Low	Low	Low	High
A22	High	Low	Low	High
A23	High	Low	Low	High
A24	Low	Low	Low	High
A25	High	Low	Low	High
A26	High	Low	Low	High
A27	Low	Low	Low	Unclear
A28	Low	Low	Low	High
A29	High	Low	Low	High
A30	Low	Low	Low	Low
A31	Low	Low	Low	Unclear
A32	High	Low	Low	High
A33	Low	Low	Low	Low
A34	Low	Low	Low	High
A35	High	Low	Low	High
A36	High	Low	Low	High
A37	Low	Low	Low	Unclear
A38	Unclear	Low	Low	High
A39	Low	Low	Low	High
A40	Unclear	Low	Low	High
A41	High	Low	Low	High
A42	Low	Low	Low	High
A43	High	Low	Low	High
A44	High	Low	Low	High
A45	High	Low	Low	High
A46	Low	Low	Low	High

## Data Availability

This study is a systematic literature review, incorporating 46 original research studies published by other scholars. All cited works are listed in the References section. Additionally, we have made all extracted data and relevant information available as Appendix A. Further inquiries can be directed to the corresponding author.

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
