# Peer review of "Applications of Artificial Intelligence for Metastatic Gastrointestinal Cancer: A Systematic Literature Review"

_cancers, 2025, doi:10.3390/cancers17030558_

Round 1
Reviewer 1 Report
Comments and Suggestions for Authors
This article provides valuable insights into the role of AI in metastatic gastrointestinal cancers and offers guidance for future research directions. By addressing the methodological challenges raised, future research can further enhance the application of AI in this field.
1. The article may need to discuss more on the application of AI models in actual clinical settings, as well as how to overcome the challenges in implementation.
2. It is recommended that the authors discuss in detail the challenges of data preprocessing and the importance of model interpretability in the discussion section, and provide specific solutions or suggestions.
3. It is suggested that the authors provide more technical details in the appendix or supplementary materials.
4. It is recommended that the authors discuss in detail the results of the PROBAST assessment in the discussion section, and provide suggestions on how to reduce the risk of bias.
5. It is suggested that the article undergo language polishing to improve the accuracy and fluency of expression. Attention should be paid to checking grammatical errors, spelling mistakes, and improper use of punctuation.
Author Response
|
Reviewer 1 |
Response |
|
This article provides valuable insights into the role of AI in metastatic gastrointestinal cancers and offers guidance for future research directions. By addressing the methodological challenges raised, future research can further enhance the application of AI in this field. 1. The article may need to discuss more on the application of AI models in actual clinical settings, as well as how to overcome the challenges in implementation. |
Thanks for your kind comment the study. In response, we have expanded the discussion to include a detailed exploration of the challenges associated with implementing AI models in clinical practice. We specifically address the importance of overcoming challenges such as data quality, standardization of preprocessing methods, and improving model interpretability to foster clinical trust. Furthermore, we provide specific suggestions for future research, including the need for prospective studies, large-scale trials, and collaboration between multidisciplinary teams to enhance the integration of AI into clinical settings. You can find mentioned information in paragraph 2-5 of discussion highlighted in green now. |
|
2. It is recommended that the authors discuss in detail the challenges of data preprocessing and the importance of model interpretability in the discussion section, and provide specific solutions or suggestions. |
Thanks for your suggestion. As recommended, we have provided a more in-depth discussion on the challenges of data preprocessing, emphasizing the need for standardized methods and handling missing data in clinical datasets. Additionally, we have elaborated on the importance of model interpretability and the development of explainable AI techniques. Specific solutions, such as the use of model-agnostic interpretation tools and the integration of domain knowledge to enhance model transparency, have been included to address these challenges. |
|
3. It is suggested that the authors provide more technical details in the appendix or supplementary materials. |
Thanks for mentioning this. Now we have uploaded the supplementary files including keywords, ROB checklist and CHARMS checklist- Please see S1, S2, and S3 files. |
|
4. It is recommended that the authors discuss in detail the results of the PROBAST assessment in the discussion section, and provide suggestions on how to reduce the risk of bias. |
Thanks for your comment. We have now incorporated an analysis of the PROBAST results in the revised manuscript. We discuss how the risk of bias was assessed in the included studies and the potential limitations these biases introduce. Additionally, we provide recommendations to mitigate such risks in future research, such as the use of prospective datasets and transparent reporting of model development processes. |
|
5. It is suggested that the article undergo language polishing to improve the accuracy and fluency of expression. Attention should be paid to checking grammatical errors, spelling mistakes, and improper use of punctuation. |
Thanks for your suggestion. We have requested MDPI language edit service to increase the fluency of language. |
Reviewer 2 Report
Comments and Suggestions for Authors
During reading manuscript I noticed that Author should include more detailed knowledge in the text with findings. here is some suggestion
1. Expand Scope of Research Analysis: The authors should provide a comprehensive analysis of AI-based published studies in the field of medical science, including applications in various types of cancer. This broader perspective would enrich the manuscript and set a comparative context. Following this, the authors should focus specifically on gastrointestinal (GI) cancer research and include a detailed literature review of relevant studies. This approach would provide a more holistic understanding of AI's potential in cancer diagnosis and treatment.
2. Summary of AI Methods with Performance Metrics: To assist clinicians and readers in understanding the clinical implications, the authors should include a summary of different AI methods applied in metastatic GI cancers. This summary should highlight each method's accuracy, sensitivity, and specificity rates. Such information would help identify areas needing improvement and recommend the most reliable approaches for practical applications. Additionally, this analysis could guide future research efforts by emphasizing the gaps and limitations of existing methodologies.
3. Reference Usage: It has been noted that several references are clustered in one place within the manuscript, which can make it difficult for readers to follow specific points or findings. The authors should distribute references more appropriately throughout the manuscript to provide clear and specific citations that support individual statements or claims. This would improve the manuscript’s clarity and readability.
Comments on the Quality of English LanguageNeed more improvement
Author Response
|
Reviewer 2 |
Response |
|
During reading manuscript I noticed that Author should include more detailed knowledge in the text with findings. here is some suggestion |
Thanks for your time reading our manuscript in detail. |
|
1. Expand Scope of Research Analysis: The authors should provide a comprehensive analysis of AI-based published studies in the field of medical science, including applications in various types of cancer. This broader perspective would enrich the manuscript and set a comparative context. Following this, the authors should focus specifically on gastrointestinal (GI) cancer research and include a detailed literature review of relevant studies. This approach would provide a more holistic understanding of AI's potential in cancer diagnosis and treatment. |
Thank you for your suggestion. During our analysis, we initially excluded studies on other types of metastatic cancers to maintain a focused scope. However, based on your recommendation, we have now included a brief description of these studies in the first paragraph of the Results section. This addition provides a broader context and highlights the comparative potential of AI applications in the diagnosis and treatment of GI and other cancers. We hope this enhancement addresses your concern and enriches the manuscript. |
|
2. Summary of AI Methods with Performance Metrics: To assist clinicians and readers in understanding the clinical implications, the authors should include a summary of different AI methods applied in metastatic GI cancers. This summary should highlight each method's accuracy, sensitivity, and specificity rates. Such information would help identify areas needing improvement and recommend the most reliable approaches for practical applications. Additionally, this analysis could guide future research efforts by emphasizing the gaps and limitations of existing methodologies. |
We appreciate the reviewer’s valuable suggestion. In response, we have added Table 6, 8, which summarizes the evaluation metrics of various AI methods. This table aims to assist clinicians and readers in understanding the clinical implications of each AI method, highlighting their strengths and areas for improvement. Furthermore, we have expanded the discussion in the manuscript to include a more detailed analysis of the limitations of current AI methodologies and provided specific suggestions for future research. These recommendations focus on addressing gaps in the existing approaches, such as the need for more diverse and larger datasets, standardized preprocessing methods, and improvements in model interpretability. We believe these additions will help guide future research efforts and promote the development of more reliable and clinically applicable AI models. |
|
3. Reference Usage: It has been noted that several references are clustered in one place within the manuscript, which can make it difficult for readers to follow specific points or findings. The authors should distribute references more appropriately throughout the manuscript to provide clear and specific citations that support individual statements or claims. This would improve the manuscript’s clarity and readability. |
Thank you for pointing this out. We have revised the manuscript by redistributing the references more evenly throughout the text. This adjustment ensures that citations are clearly associated with specific statements or findings, making it easier for readers to follow and understand the information presented. |
Round 2
Reviewer 1 Report
Comments and Suggestions for Authors
The author has addressed my concerns.